# Patterns of routine primary care for osteoarthritis in the UK: a cross-sectional electronic health records study

Holly Jackson,[1,2] Lauren A Barnett,[1] Kelvin P Jordan,[1,3] Krysia S Dziedzic,[1] Elizabeth Cottrell,[1] Andrew G Finney,[1] Zoe Paskins,[1,4] John J Edwards[1]

[1]Arthritis Research UK Primary Care Centre, Keele University Research Institute for Primary Care and Health Sciences, Keele, UK
[2]School of Mathematical Sciences, University of Nottingham, Nottingham, UK
[3]Keele Clinical Trials Unit, Keele University, Staffordshire, UK
[4]Rheumatology Department, The Haywood Hospital, Stoke-on-Trent, UK

**Correspondence to**
Dr John J Edwards;
j.j.edwards@keele.ac.uk

## ABSTRACT

**Objective** To determine common patterns of recorded primary care for osteoarthritis (OA), and patient and provider characteristics associated with the quality of recorded care.

**Design** An observational study nested within a cluster-randomised controlled trial.

**Setting** Eight UK general practices who were part of the Management of Osteoarthritis in Consultations study.

**Participants** Patients recorded as consulting within the eight general practices for clinical OA.

**Primary outcomes** Achievement of seven quality indicators of care (pain/function assessment, information provision, exercise/weight advice, analgesics, physiotherapy), recorded through an electronic template or routinely recorded in the electronic healthcare records, was identified for patients aged ≥45 years consulting over a 6-month period with clinical OA. Latent class analysis was used to cluster patients based on care received. Clusters were compared on patient and clinician-level characteristics.

**Results** 1724 patients (median by practice 183) consulted with clinical OA. Common patterns of recorded quality care were: cluster 1 (38%, *High*) received most quality indicators of care; cluster 2 (11%, *Moderate*) had pain and function assessment, and received or were considered for other indicators; cluster 3 (17%, *Low*) had pain and function assessment, and received or were considered for paracetamol or topical non-steroidal anti-inflammatory drugs; cluster 4 (35%, *None*) had no recorded quality indicators. Patients with higher levels of recorded care consulted a clinician who saw more patients with OA, consulted multiple times and had less morbidity. Those in the *High* cluster were more likely to have recorded diagnosed OA and have knee/hip OA.

**Conclusions** Patterns of recorded care for OA fell into four natural clusters. Appropriate delivery of core interventions and relatively safe pharmacological options for OA are still not consistently recorded as provided in primary care. Further research to understand clinical recording behaviours and determine potential barriers to quality care alongside effective training for clinicians is needed.

**Trial registration number** ISRCTN06984617; Results.

### Strengths and limitations of this study

► This paper describes a novel use of latent class analysis to identify patterns of primary care for osteoarthritis (OA).
► The population studied was large and diverse, increasing generalisability, and based on a broad definition of clinical OA to reduce selection bias.
► The analysis used some quality indicators of care newly implemented in practices through an electronic template (pain/function assessment, information provision, exercise/weight advice, analgesics, physiotherapy), which may have increased the recorded quality of care compared with routine practice.
► Four clusters of recorded care were identified: approximately one-third of patients had a high probability of delivery of most care processes while another third had a low probability of any such delivery. The remaining patients had a high probability of pain and function assessment but were distinguished by the probability of delivery or consideration of other aspects of care.

## INTRODUCTION

Osteoarthritis (OA) is a common reason for adults aged ≥45 years to consult primary care. Annually, in the UK, 4% of such adults are recorded as consulting in general practice for diagnosed OA, with an additional 8% recorded with joint pain likely to be attributable to OA.[1] OA is a common reason for disability, and was ranked the 11th biggest cause of disability by the 2010 Global Burden of Disease study.[2]

The UK National Institute for Health and Care Excellence (NICE) OA management guidelines recommend core strategies of information provision, physical activity and exercise, and weight management, supplemented with use of relatively safe pharmacological management strategies (eg, topical non-steroidal anti-inflammatory drugs (NSAID)), as necessary.[3] Intensification of management should depend on response to these initial approaches. However, there is evidence that patients diagnosed with OA do not receive care that

is well aligned to evidence-based recommendations and which may be overly dependent on pharmacological methods.[4]

We have previously identified variation between clinicians in recorded quality of individual indicators of OA care.[5] However, patterns of OA care and factors linked with increased probability of adherence to OA quality standards are less well studied. Using electronic general practice records data, the objectives of this study were to determine patterns of recorded primary care for OA based on quality indicators, and to determine associations between higher quality recorded care and patient and clinician characteristics.

## METHODS

This analysis used data from the Management of Osteoarthritis in Consultations (MOSAICS) study (trial registration number ISRCTN06984617).[6] MOSAICS was a mixed-methods study, which investigated the effect of a model consultation for clinical OA. It was set within eight general practices in Cheshire, Shropshire and Staffordshire, UK. Practice eligibility has been reported elsewhere.[6] The current analysis, reported in line with Strengthening the Reporting of Observational Studies in Epidemiology guidelines, used anonymised information from the electronic health records (EHR) of these practices for the 6-month baseline period before randomisation of practices to intervention or control arms.[6] At the beginning of the baseline period, a computerised template ('e-template', described below) was installed within the EHR and all practices continued with otherwise usual care until the end of the baseline period.

The study population was all patients aged ≥45 years registered with the eight general practices who consulted with clinical OA in the baseline 6-month period. UK general practice uses a system of Read codes (similar in principle to the International Classification of Diseases codes) to record symptoms, morbidities and care processes[7]; within MOSAICS, clinical OA was defined as either a recorded OA Read code or a peripheral joint pain Read code for the hand, hip, knee or foot, to reduce the potential for selection bias in clinician coding. Patients were allocated to an index clinician, being the clinician recording the first formally diagnosed (ie, OA Read-coded) OA consultation in the baseline period or, if none, the first peripheral joint pain coded consultation in the same period.

Outcome measures were the seven indicators of quality of care for OA in general practice recorded in the EHR (table 1). These could be entered into the EHR as routinely recorded data or captured through the e-template. The identification and synthesis of appropriate quality indicators using a systematic review and NICE 2008 guidelines has previously been reported.[5 8 9]

**Table 1** Seven quality Indicators and categories used for latent class analysis

| Quality indicator | Categories | Definition |
|---|---|---|
| 1. Pain assessed | Assessed<br>Not assessed | Recorded level of pain*<br>No entry recorded* |
| 2. Function assessed | Assessed<br>Not assessed | Recorded level of function*<br>No entry recorded* |
| 3. OA information | Given<br>Considered, but not given<br>Not considered | Recorded written or verbal*<br>Recorded not appropriate*<br>No entry recorded* |
| 4. Exercise advice | Given<br>Considered, but not given<br>Not considered | Recorded written or verbal*<br>Recorded not appropriate*<br>No entry recorded* |
| 5. Weight loss advice† | Given<br>Considered, but not given<br>Not considered | Recorded written or verbal*<br>Recorded not appropriate*<br>No entry recorded* |
| 6. Paracetamol or topical NSAID | Prescribed<br>Considered, but not prescribed<br>Not considered | Either drug prescribed‡<br>Neither drug prescribed but recorded tried, offered, patient declined, or not appropriate*<br>Neither drug prescribed, recorded unknown or no entry recorded for both drugs* |
| 7. Physiotherapy | Referred<br>Considered, but not referred<br>Not considered | Recorded referral‡<br>No referral but recorded as offered, or not necessary or not appropriate*<br>No referral, recorded not this time or no entry recorded* |

*From e-template.
†Patients without a recorded BMI of ≥25 within the last 3 years were allocated to 'Considered, but not given' category.
‡From routine records.
BMI, body mass index; NSAID, non-steroidal anti-inflammatory drug; OA, osteoarthritis.

Achievement of prescribing and referral indicators (recorded prescription of topical NSAIDs or paracetamol, and onward physiotherapy referral) were determined from data in the routinely recorded component of the EHR and were determined to have been achieved if they were recorded within 14 days of any clinical OA consultation in the 6-month period.

The e-template facilitated recording of achievement of indicators that are known to be poorly captured in routinely recorded data[5]: (1) assessment of pain and function; (2) provision or consideration of OA information, exercise advice and weight loss advice; (3) consideration of paracetamol or topical NSAID; and (4) consideration of physiotherapy referral. The entry of a code for clinical OA for a patient aged ≥45 years triggered the e-template. The design, effects, and interpretation of the e-template have previously been reported.[5] The clinicians could complete the e-template at any point throughout the consultation and could choose to complete all, some or none of the e-template. The e-template has been endorsed by NICE to facilitate enhanced uptake of quality standards.[10]

Data from the EHR (derived from both routinely recorded data and the e-template) were amalgamated within the relevant quality indicator. For example, consideration of paracetamol and topical NSAIDs (entered using e-template) was combined with actual prescription of these agents (routinely recorded data). Outcomes (table 1) were dichotomous for pain and function assessments. For all other indicators, the possibilities were for the indicator to be *achieved, considered* (without record of having been delivered) or *not considered.* There is evidence that weight recording is more common in people who are overweight compared with those who are not.[11] To minimise the effect of missing data and to preserve the ability of the model to identify people who needed weight loss advice but were not recorded as receiving it, any patient recorded as being of normal weight or who did not have a weight recorded was allocated to *considered* for weight loss advice.

We investigated how patterns of care based on the quality indicators were associated with other OA care processes, recorded in the routine EHR within 14 days of any clinical OA consultation: prescriptions for oral NSAIDs and opioids, and relevant X-rays (hand, hip, knee or foot).

Factors potentially associated with patterns of quality of care that were considered were: patient age, gender, body mass index, the site of clinical OA, whether patients had multiple or a single consultation for clinical OA within the 6-month time period, whether the patient was a new consulter (no clinical OA consultations within the previous 12 months) and total morbidity. Total morbidity was measured by a count of British National Formulary subchapters from which prescriptions had been issued in the previous 12 months.[12] A proxy measure of OA workload for the patients' index clinician was determined by dichotomising the number of index clinical OA consultations at the median value (14) across clinicians.

## Statistical analysis

Latent class analysis (LCA) was used to cluster patients into groups based on recorded achievement of the seven quality indicators. All patients within a cluster should have similar recorded care for their OA or joint pain, but care should differ between patients belonging to different clusters.[13]

Latent class models were fitted, beginning with a one-cluster model where all the patients were assumed to have been given the same pattern of treatment of OA, up to a seven-cluster model. To determine the optimum number of clusters, we considered the Bayes information criterion[14] (BIC, where the lowest BIC indicated the best model) with the size of each cluster, and the interpretability of the model. Posterior probabilities (PP) for a patient (the probabilities of that patient belonging to each of the clusters within the model) were identified. The cluster that had the largest PP for a patient was the cluster that patient was assigned to. We used the mean PP for patients allocated to each cluster to measure cluster separation; a mean PP of more than 0.7 indicated that the patients were clearly assigned to that specific cluster.[15]

Using a two-level (patient within index clinician) multinomial multilevel logistic regression, associations between the patient and clinician-level covariates and cluster membership were estimated and reported as relative risk ratios (RRR) with 95% CIs. We also used $X^2$ tests to compare between clusters on levels of pain and functional limitation (none, mild, moderate, severe) as recorded in the e-template.

Statistical analysis was performed using R studio V.3.3.0, and MLwiN V.2.35 for Windows.

## RESULTS

During the 6-month period, 1724 patients (median per practice n=183) consulted with a recorded clinical OA code and triggered the e-template. All were included in the analysis. 1014 (59%) of these were female, mean age was 66.1 years (SD: 11.9) and 582 (34%) patients were recorded with a diagnosis of OA rather than peripheral joint pain. Among consulters, 50% were recorded as having clinical OA at the knee, 21% at the hip, and the remainder with ankle/foot, wrist/hand, multisite, or unspecified clinical OA.

As previously reported,[5] pain (63%) and function (62%) assessments were the most commonly achieved indicators. Recorded provision of OA information (44%) and exercise advice (45%) were achieved in under half of patients, and weight loss advice in less than a third of patients (31%). Up to 609 (35%) patients were prescribed paracetamol or topical NSAIDs. A referral for physiotherapy was made in 7% of patients.

Table 2 shows the goodness-of-fit statistics for the LCA models with one to seven clusters. The four-cluster model gave the lowest BIC, and each of the clusters in the three, four and five-cluster models had a mean PP for patients belonging to that cluster above 0.83. In the three-cluster

**Table 2** Latent class analysis of goodness-of-fit statistics

| Number of clusters | BIC | $x^2$ goodness of fit | Population (%) of smallest cluster | Range of mean PP across clusters | n (%) with PP<0.7 |
|---|---|---|---|---|---|
| 1 | 20 994.14 | 32 978.08 | 1724 (100) | 1.000 | 0 (0) |
| 2 | 15 160.57 | 3332.77 | 1071 (62) | 0.992, 0.987 | 3 (<1) |
| 3 | 14 715.82 | 1727.74 | 430 (25) | 0.906, 0.991 | 138 (8) |
| 4 | 14 627.48 | 1522.28 | 184 (11) | 0.848, 0.994 | 157 (9) |
| 5 | 14 661.55 | 809.88 | 142 (8) | 0.830, 0.993 | 207 (12) |
| 6 | 14 699.79 | 733.23 | 112 (6) | 0.754, 0.996 | 257 (15) |
| 7 | 14 771.09 | 818.78 | 22 (1) | 0.701, 0.996 | 267 (15) |

BIC, Bayes information criterion; PP, posterior probabilities.

model, the smallest cluster size was 430 (25%), in the four-cluster model it was 184 (11%), and the five-cluster model had the smallest cluster size of 142 (8%). Based on the cluster sizes, goodness-of-fit statistics and clinical interpretability, the four-cluster model was chosen as the optimal model.

Table 3 shows the probability of recorded receipt of each of the seven quality indicators for patients allocated to each cluster. Patients in cluster 1 (n=659, 38%) had a high probability of having pain and function assessment recorded (probabilities over 0.97) and of being given OA information and exercise advice (probabilities over 0.93). Patients' care within this cluster was recorded as having achieved a median of five indicators and considered for, but not achieved, a median of one further indicator. Cluster 1 was therefore labelled as having a *High* level of recorded quality of care. Cluster 2 (n=184, 11%; *Moderate*) had a high probability of pain and function assessment (probabilities over 0.95) and of consideration for (but not receipt of) physiotherapy and topical NSAID

**Table 3** Conditional item response probabilities for the quality indicators for each cluster

| | Quality indicators | Overall n (%) | Cluster High (n=659, 38%) | Moderate (n=184, 11%) | Low (n=286, 17%) | None (n=595, 35%) |
|---|---|---|---|---|---|---|
| Pain assessment | Assessed | 1092 (63) | 0.978 | 0.961 | 0.922 | 0.014 |
| | Not assessed | 632 (37) | 0.022 | 0.039 | 0.078 | 0.987 |
| Function assessment | Assessed | 1070 (62) | 0.981 | 0.955 | 0.873 | 0.000 |
| | Not assessed | 654 (38) | 0.019 | 0.045 | 0.127 | 1.000 |
| OA information | Given | 764 (44) | 0.930 | 0.463 | 0.319 | 0.001 |
| | Considered, not given | 85 (5) | 0.009 | 0.330 | 0.011 | 0.000 |
| | Not considered | 875 (51) | 0.062 | 0.207 | 0.670 | 1.000 |
| Exercise advice | Given | 768 (45) | 0.994 | 0.417 | 0.237 | 0.000 |
| | Considered, not given | 96 (6) | 0.007 | 0.313 | 0.067 | 0.000 |
| | Not considered | 860 (50) | 0.000 | 0.270 | 0.696 | 1.000 |
| Weight advice | Given | 536 (31) | 0.593 | 0.115 | 0.089 | 0.000 |
| | Considered, not given | 153 (9) | 0.298 | 0.733 | 0.347 | 0.441 |
| | Not considered | 1035 (60) | 0.109 | 0.152 | 0.564 | 0.559 |
| Topical NSAID/ paracetamol | Prescribed | 609 (35) | 0.476 | 0.273 | 0.394 | 0.239 |
| | Considered, not prescribed | 570 (33) | 0.496 | 0.641 | 0.406 | 0.004 |
| | Not considered | 545 (32) | 0.028 | 0.086 | 0.200 | 0.757 |
| | Referred | 124 (7) | 0.111 | 0.037 | 0.101 | 0.032 |
| Physiotherapy | Considered, not referred | 532 (31) | 0.559 | 0.732 | 0.080 | 0.000 |
| | Not considered | 1068 (62) | 0.330 | 0.230 | 0.819 | 0.968 |
| Median count (IQR): assessed/prescribed/given/referred | | | 5 (4. 6) | 3 (2. 3) | 3 (2. 3) | 0 (0. 1) |
| Median count (IQR): considered | | | 1 (1. 2) | 3 (2. 4) | 1 (0. 1) | 0 (0. 1) |

NSAID, non-steroidal anti-inflammatory drug; OA, osteoarthritis.

or paracetamol. They also had a high probability of being given or considered for OA information and exercise advice. Their recorded care achieved a median of three indicators and they were considered for care relating to a median of three further indicators. Cluster 3 (n=286, 17%; *Low*) had a high probability of pain and function assessment (probabilities over 0.87), and was likely to be prescribed or considered for paracetamol or topical NSAIDs but generally was not recorded as receiving or being considered for other indicators (received a median of three processes and considered for a median of one further). Cluster 4 (n=595, 35%; *None*) had low probabilities of a record of receiving or being considered for any indicator (received and considered median zero indicators).

Table 4 compares the number of people in each cluster who were expected, based on the model, to receive each care process (identified by the indicators) and the number actually recorded as receiving them. Differences between observed and expected values were small and generally related to distinguishing between care received compared with care considered. For example, in the pain assessment domain, there was no difference between the counts of observed and expected provision for the *High* and *Moderate* clusters, and a difference of only one patient in the *Low* and *None* clusters; for OA information provision, this was observed more frequently than expected for the *High* cluster (observed n=620 compared with 613 expected) but less frequently for the *Moderate* (59 vs 85) and *Low* (85 vs 91) clusters.

Patient and clinician characteristics for each cluster are shown in table 5 with results from the multinomial model comparing clusters in table 6. Compared with the *None* cluster, patients in the *High* and *Moderate* clusters tended to consult with a clinician with a higher OA workload, consult multiple times and have less total morbidity (table 6). The patients with *High* level of recorded care were more likely to have diagnosed OA (adjusted RRR 1.81, 95% CI 1.41 to 2.32) and less likely to have hand or foot clinical OA than patients in the *None* cluster, while patients in the *Moderate* cluster were less likely to have diagnosed OA (RRR 0.55, 95% CI 0.35 to 0.85) or be overweight (RRR 0.57, 95% CI 0.39 to 0.85), but more likely to have clinical OA in multiple sites (RRR 1.89, 95% CI 0.99 to 3.59) than patients in the *None* cluster. Patients in the *Low* cluster were less likely than patients in the *None* cluster to have a single consultation (RRR 0.45, 95% CI 0.34 to 0.60), have clinical OA in the foot (RRR 0.25, 95% CI 0.13 to 0.51) or have multimorbidity.

Those in the *High* cluster had slightly higher levels of opioid prescription (36%; $X^2$ test, P=0.06), oral NSAID prescription (20%; P=0.01) and recorded X-rays (22%; P<0.01) than patients in the other clusters, although differences between the *High* and *Low* clusters, in particular, were small (table 7).

In those with a record of a pain assessment, patients in the *High* cluster were more likely to have recorded moderate or severe pain (70% vs 57% in the *Moderate*

cluster and 64% in the *Low* cluster). The same pattern was seen for functional limitation although differences between clusters were smaller (table 7).

## DISCUSSION

This study has identified four patterns of recorded primary care management of OA based on previously identified quality indicators of care. Just over a third of patients consulting for clinical OA had recorded care meeting the majority of quality indicators. Another third were not recorded as having received or been considered for any of these quality indicators. Factors associated with higher recorded quality of care included receiving an OA diagnosis, OA in the knee or hip rather than foot or hand, lower total morbidity burden, multiple consultations for clinical OA, and initial consultation with a clinician who was recorded as seeing more than the median number of patients with OA. Previous evidence has demonstrated that guidelines for treatment of OA within primary care are not consistently adhered to.[16–18] The way in which receipt of different recommended care processes for OA are grouped within patients has not previously been investigated. In our study, 38% of the patients were recorded as having received a relatively large number of quality indicators and could be regarded as a group achieving the closest to optimal care based on these indicators (the *High* group). Care for members of two clusters (*Moderate* and *Low*) achieved some quality indicators overall but can be distinguished by the fact that information, advice (exercise, weight loss) and physiotherapy were more likely to be considered in the *Moderate* cluster than the *Low* cluster. A third of patients were in the *None* cluster which demonstrated the weakest recorded quality of care with the majority of this group lacking recorded achievement or consideration of any indicator. The patients in the cluster with the best recorded care (*High*) were also more likely to receive other elements of care such as oral NSAIDs and referral for X-ray. NICE does not recommend routine use of X-ray for OA diagnosis and suggests that opioids and oral NSAIDS should be used only if topical NSAIDs and paracetamol do not relieve pain.[3] The greater use of these approaches in the *High* cluster may reflect worse severity of OA and this cluster did have slightly higher levels of clinician-recorded pain and functional limitation than those in the *Moderate* and *Low* clusters. While one hypothesis may be that patients in the *High* cluster are given all possible care elements, this is unlikely to be the case as differences between clusters on the non-quality indicator elements of care were generally small, and most patients in the *High* cluster were not in receipt of these non-recommended approaches.

It is possible that the clinicians treating those in the *High* cluster were more engaged with, or more confident in managing OA. Confidence in OA management could be associated with confidence in OA diagnosis, which may explain the increased use of OA Read codes in these patients. Conversely, where OA Read codes were

**Table 4** Expected number compared with observed for each category of indicators, by cluster

| Quality indicators | | Cluster | | | | | | | | | | | |
| | | High (n=659, 38%) | | | Moderate (n=184, 11%) | | | Low (n=286, 17%) | | | None (n=595, 35%) | | |
| | | E | O | Δ | E | O | Δ | E | O | Δ | E | O | Δ |
| Pain assessment | Assessed (n=1092, 63%) | 645 | 645 | 0 | 177 | 177 | 0 | 264 | 263 | 1 | 8 | 7 | 1 |
| | Not assessed (n=632, 37%) | 14 | 14 | 0 | 7 | 7 | 0 | 22 | 23 | −1 | 587 | 588 | −1 |
| Function assessment | Assessed (n=1070, 62%) | 646 | 646 | 0 | 176 | 174 | 2 | 250 | 250 | 0 | 0 | 0 | 0 |
| | Not assessed (n=655, 38%) | 13 | 13 | 0 | 8 | 10 | −2 | 36 | 36 | 0 | 595 | 595 | 0 |
| OA information | Given (n=764, 44%) | 613 | 620 | −7 | 85 | 59 | 26 | 91 | 85 | 6 | 0 | 0 | 0 |
| | Considered, *not* given (n=85, 5%) | 6 | 3 | 3 | 61 | 81 | −20 | 3 | 1 | 2 | 0 | 0 | 0 |
| | Not considered (n=875, 51%) | 41 | 36 | 5 | 38 | 44 | −6 | 192 | 200 | −8 | 595 | 595 | 0 |
| Exercise advice | Given (n=768, 45%) | 655 | 658 | −3 | 77 | 53 | 24 | 68 | 57 | 11 | 0 | 0 | 0 |
| | Considered, *not* given (n=96, 6%) | 4 | 1 | 3 | 58 | 77 | −19 | 19 | 18 | 1 | 0 | 0 | 0 |
| | Not considered (n=860, 50%) | 0 | 0 | 0 | 50 | 54 | −4 | 199 | 211 | −12 | 595 | 595 | 0 |
| Weight advice | Given (n=536, 31%) | 391 | 370 | 21 | 21 | 20 | 1 | 26 | 22 | 4 | 0 | 0 | 0 |
| | Considered, *not* given (n=153, 9%) | 196 | 213 | −17 | 135 | 140 | −5 | 99 | 99 | 0 | 262 | 262 | 0 |
| | Not considered (n=1035, 60%) | 72 | 76 | −4 | 28 | 24 | 4 | 161 | 165 | −4 | 333 | 333 | 0 |
| Topical NSAID or paracetamol | Prescribed (n=609, 35%) | 314 | 311 | 3 | 50 | 47 | 3 | 113 | 111 | 2 | 142 | 140 | 2 |
| | Considered, *not* prescribed (n=570, 33%) | 327 | 330 | −3 | 118 | 119 | −1 | 116 | 118 | −2 | 2 | 3 | −1 |
| | Not considered (n=545, 32%) | 18 | 18 | 0 | 16 | 18 | −2 | 57 | 57 | 0 | 450 | 452 | −2 |
| Physiotherapy | Referred (n=124, 7%) | 73 | 69 | 4 | 7 | 6 | 1 | 29 | 30 | −1 | 19 | 19 | 0 |
| | Considered, *not* referred (n=532, 31%) | 369 | 371 | −2 | 135 | 147 | −12 | 23 | 14 | 9 | 0 | 0 | 0 |
| | Not considered (n=1068, 62%) | 218 | 219 | −1 | 42 | 31 | 11 | 234 | 242 | −8 | 576 | 576 | 0 |

E, expected number; NSAID, non-steroidal anti-inflammatory drug; O, observed number; OA, osteoarthritis; Δ, difference.

**Table 5** Patient and clinician characteristics for each cluster

| | | Cluster | | | |
|---|---|---|---|---|---|
| | Total n (%) | High (n=659) | Moderate (n=18) | Low (n =286) | None (n=595) |
| **Patient factors** | | | | | |
| Age | | | | | |
| 45–64 | 817 | 277 (34) | 109 (13) | 293 (43) | 138 (17) |
| 65–74 | 442 | 213 (48) | 20 (5) | 144 (33) | 65 (15) |
| 75–84 | 349 | 133 (38) | 35 (10) | 116 (33) | 65 (19) |
| 85+ | 116 | 36 (31) | 20 (17) | 42 (36) | 18 (6) |
| Gender | | | | | |
| Male | 710 | 286 (40) | 68 (10) | 113 (16) | 243 (34) |
| Female | 1014 | 373 (37) | 116 (11) | 173 (17) | 352 (35) |
| BMI category | | | | | |
| Normal | 315 | 111 (35) | 54 (17) | 48 (15) | 102 (32) |
| Overweight | 1080 | 471 (44) | 83 (8) | 193 (18) | 333 (31) |
| Not recorded | 329 | 77 (23) | 47 (14) | 45 (14) | 160 (49) |
| Diagnosis | | | | | |
| Recorded with joint pain only | 1142 | 366 (32) | 148 (13) | 207 (18) | 421 (37) |
| OA diagnosis | 582 | 293 (50) | 36 (6) | 79 (14) | 174 (30) |
| Site of OA | | | | | |
| Knee | 855 | 359 (42) | 80 (9) | 149 (17) | 267 (31) |
| Hip | 363 | 135 (37) | 41 (11) | 68 (19) | 119 (33) |
| Foot | 125 | 30 (24) | 15 (12) | 10 (8) | 70 (56) |
| Hand | 152 | 33 (22) | 25 (16) | 31 (20) | 63 (41) |
| Unspecified | 99 | 30 (30) | 8 (8) | 16 (16) | 45 (46) |
| Multiple | 130 | 72 (55) | 15 (12) | 12 (9) | 31 (24) |
| Morbidity load* | | | | | |
| BNF count 0–4 | 485 | 156 (32) | 68 (14) | 89 (18) | 172 (36) |
| 5–9 | 578 | 240 (42) | 56 (10) | 99 (17) | 183 (32) |
| 10+ | 661 | 263 (40) | 60 (9) | 98 (15) | 240 (36) |
| Number of OA consultations† | | | | | |
| Multiple | 532 | 250 (47) | 63 (12) | 99 (19) | 120 (23) |
| Single | 1192 | 409 (34) | 121 (10) | 187 (16) | 475 (40) |
| Median (IQR) number of OA consultations† | 1 (0, 1) | 1 (1, 2) | 1 (1, 2) | 1 (1, 2) | 1 (1, 1) |
| Consulter status | | | | | |
| Repeat | 566 | 232 (41) | 53 (9) | 84 (15) | 197 (35) |
| New‡ | 1158 | 427 (37) | 131 (11) | 202 (17) | 398 (34) |
| **Clinician factors** | | | | | |
| Clinician OA workload† | | | | | |
| Below the median | 197 | 41 (21) | 16 (8) | 36 (18) | 104 (53) |
| Above the median | 1527 | 618 (41) | 168 (11) | 250 (16) | 491 (32) |

*Number of BNF subchapters from which prescription was made in previous 12 months.
†During 6-month period.
‡No clinical OA consultations within the previous 12 months.
BMI, body mass index; BNF, British National Formulary; OA, osteoarthritis.

**Table 6**  Associations of patient and clinician characteristics with cluster membership

| n=1724 | High versus None RRR* (95% CI) | Moderate versus None RRR* (95% CI) | Low versus None RRR* (95% CI) |
|---|---|---|---|
| **Patient factors** | | | |
| **Age** | | | |
| 45–64 | 1 | 1 | 1 |
| 65–74 | 1.41 (1.07 to 1.84) | 0.45 (0.27 to 0.74) | 0.97 (0.69 to 1.37) |
| 75–84 | 1.13 (0.83 to 1.52) | 1.02 (0.65 to 1.60) | 1.42 (0.99 to 2.05) |
| 85+ | 0.91 (0.56 to 1.47) | 1.56 (0.85 to 2.89) | 1.24 (0.69 to 2.23) |
| **Gender** | | | |
| Male | 1 | 1 | 1 |
| Female | 0.86 (0.69 to 1.07) | 1.03 (0.75 to 1.43) | 1.04 (0.80 to 1.36) |
| **BMI category** | | | |
| Normal | 1 | 1 | 1 |
| Overweight | 1.20 (0.91 to 1.60) | 0.57 (0.39 to 0.85) | 1.33 (0.93 to 1.90) |
| Not recorded | 0.39 (0.27 to 0.56) | 0.52 (0.33 to 0.81) | 0.52 (0.33 to 0.82) |
| **Diagnosis** | | | |
| Recorded with joint pain only | 1 | 1 | 1 |
| OA diagnosis | 1.81 (1.41 to 2.32) | 0.55 (0.35 to 0.85) | 0.93 (0.68 to 1.29) |
| **Site of OA** | | | |
| Knee | 1 | 1 | 1 |
| Hip | 0.86 (0.66 to 1.14) | 1.14 (0.76 to 1.71) | 1.04 (0.75 to 1.44) |
| Foot | 0.38 (0.24 to 0.60) | 0.73 (0.39 to 1.36) | 0.25 (0.13 to 0.51) |
| Hand | 0.45 (0.30 to 0.70) | 1.18 (0.70 to 1.98) | 0.88 (0.56 to 1.39) |
| Unspecified | 0.48 (0.30 to 0.80) | 0.85 (0.38 to 1.90) | 0.74 (0.41 to 1.34) |
| Multiple | 1.13 (0.75 to 1.74) | 1.89 (0.99 to 3.59) | 0.65 (0.34 to 1.24) |
| **Morbidity load†** | | | |
| **BNF count** | | | |
| 0–4 | 1 | 1 | 1 |
| 5–9 | 0.95 (0.71 to 1.26) | 0.74 (0.50 to 1.11) | 0.75 (0.54 to 1.06) |
| 10+ | 0.64 (0.47 to 0.87) | 0.55 (0.35 to 0.86) | 0.50 (0.35 to 0.73) |
| **Number of OA consultations‡** | | | |
| Multiple | 1 | 1 | 1 |
| Single | 0.43 (0.34 to 0.54) | 0.47 (0.33 to 0.66) | 0.45 (0.34 to 0.60) |
| **Consulter status** | | | |
| Repeat | 1 | 1 | 1 |
| New§ | 1.12 (0.89 to 1.41) | 1.09 (0.76 to 1.55) | 1.18 (0.88 to 1.59) |
| **Clinician factors** | | | |
| **Clinician OA workload‡** | | | |
| Below the median | 1 | 1 | 1 |
| Above the median | 2.90 (1.98 to 4.25) | 2.32 (1.33 to 4.03) | 1.46 (0.98 to 2.18) |

*Relative risk ratio from multilevel multinomial regression (patients within initial clinician seen) adjusted for all presented covariates, *None* cluster is reference.

†Number of BNF subchapters from which prescription was made in previous 12 months.

‡During 6-month period.

§No clinical OA consultations within the previous 12 months.

BMI, body mass index; BNF, British National Formulary; OA, osteoarthritis; RRR, relative risk ratios.

**Table 7** Use of management processes other than those used as quality indicators, and recorded severity of pain and functional limitation, by cluster

| n (column %) | Total n (%) | Cluster | | | | P value* |
| | | High (n=659) | Moderate (n=184) | Low (n=286) | None (n=595) | |
|---|---|---|---|---|---|---|
| Opioid prescribed | 557 (33) | 236 (36) | 54 (29) | 94 (33) | 173 (29) | 0.06 |
| Oral NSAID prescribed | 284 (17) | 130 (20) | 21 (11) | 49 (17) | 84 (14) | 0.01 |
| X-ray requested | 263 (15) | 142 (22) | 30 (16) | 52 (18) | 39 (7) | <0.01 |
| n with pain record | 1092 | 645 | 177 | 263 | 7 | 0.001† |
| No pain | 16 (1) | 4 (<1) | 7 (4) | 4 (2) | 1 | |
| Mild pain | 348 (32) | 187 (29) | 69 (39) | 91 (35) | 1 | |
| Moderate pain | 582 (53) | 357 (55) | 84 (47) | 136 (52) | 5 | |
| Severe pain | 146 (13) | 97 (15) | 17 (10) | 32 (12) | 0 | |
| n with function record | 1070 | 646 | 174 | 250 | 0 | 0.004† |
| No limitation | 101 (9) | 46 (7) | 29 (16) | 26 (10) | 0 | |
| Mild limitation | 456 (43) | 276 (43) | 73 (42) | 107 (43) | 0 | |
| Moderate limitation | 427 (40) | 277 (43) | 57 (33) | 93 (37) | 0 | |
| Severe limitation | 86 (8) | 47 (7) | 15 (9) | 24 (10) | 0 | |

*$\chi^2$ test.
†Excluding *None* cluster.
NSAID, non-steroidal anti-inflammatory drug.

not given there may have been uncertainty about both diagnosis and management. Previous qualitative observational research of primary care consultations has identified confusion about the construct of OA, with family doctors tending not to use the term 'osteoarthritis' with patients but instead, normalising symptoms.[19] A formal diagnosis of OA, delivered explicitly, may be needed for holistic components of care such as patient education and self-management support to be offered.[5 19] Patients with greater morbidity received a lower recorded quality of care and this may be because they were (perhaps erroneously) considered less suitable for non-pharmacological and relatively safe pharmacological options. It is also possible that OA was given lower priority compared with their other problems.[19 20] Patients with foot (and to some extent hand) OA may also have been particularly susceptible to lower levels of recorded quality of care and this site has been less well investigated with regard to effective interventions.[21 22]

This is the first study known to the authors which examines patterns of quality of care of chronic conditions such as OA. Other analyses of recorded quality of care for OA have reported some influences on individual process measures. Broadbent *et al* identified older age as being associated with reduced information provision but increased initial use of paracetamol and, where an oral NSAID was prescribed, greater first use of ibuprofen or a cyclooxygenase-2 selective NSAID; female sex was associated with increased information provision; severe OA was associated with increased pain and function assessment in the previous year.[23] Unlike in this analysis, Min *et al* identified an association between multimorbidity (using

a count of conditions) and better quality of care among vulnerable elders, some of whom had OA.[24]

This study has important strengths. The study population was large and the practices were diverse with respect to urbanisation, staffing, deprivation and size of registered population, implying good generalisability. Prescription recording is likely to be near complete since most prescribing is electronic and use of the e-template mitigates against missing data from patients using over-the-counter pharmacological approaches. The e-template also facilitates enhanced data collection in general practice without incurring biases such as social desirability. LCA uses probabilistic modelling and finite mixture distributions to collect participants into clusters, which is a different method from traditional clustering techniques (eg, cluster analysis). Given this, LCA should produce a lower misclassification rate and better statistical criteria for investigating model fit.[25] While there was variation in quality of care between clinicians and practices,[5] clustering effects of patients within clinicians were adjusted for through the multilevel model. There are some limitations in this analysis. Due to the inherent nature of EHR studies, the data extracted are a function of both the individual clinician's clinical and recording behaviours. It is therefore possible that some patients were misclassified as the lack of a record of a care process does not conclusively demonstrate that it did not occur. Compared with prescription recording, it is less certain how well-recorded referrals are. However, despite the limitations of EHR data, the differences in levels of prescribed analgesia between the clusters suggest there were real differences in care between the four clusters identified. Conversely,

patients may have been coded as receiving some elements of care without this necessarily having been conducted in a comprehensive or meaningful way. Triangulation of medical record indicators with patient-reported indicators would be needed to evaluate this further. Our assumption that those without a weight recorded were considered for weight loss advice was based on the increased likelihood of a weight recording if a patient appears overweight[11] but will have overestimated the proportion of patients considered for weight loss advice. However, over 80% of patients did have a weight record. The association between multiple consultations for OA and clusters with higher recorded quality of care may reflect greater opportunity to provide and record care but may also have reflected a greater disease severity and healthcare need. Although we considered comorbidities, previous research has identified that OA may be discussed in complex consultations about multiple problems[19] and the length of time discussing OA in a consultation would likely be an important influence on the level of recorded care. It is also possible that those with recorded peripheral joint pain rather than recorded OA may not have OA, particularly in the foot.[26] The e-template itself was previously found to be associated with increased prescription of paracetamol and topical NSAIDs and so the patterns of care recorded may not be generalisable to practices not using the e-template.[5]

Promotion of core interventions (information, exercise and weight loss advice), alongside appropriate use of the relatively safe pharmacological options, remains an important strategy in the primary care management of OA, but many patients receive few or none of these. This is particularly true for patients with higher levels of morbidity, or hand or foot OA. While there is substantial variation in recorded care of OA, high-quality care appears feasible given we found that over a third of patients with OA were recorded as receiving most core recommendations. A lack of a systematic approach to people with OA has previously been reported.[27] A structured annual review for people with OA[28] as recommended by NICE[10] may help. This may possibly be nurse led and integrated, where appropriate, into a multimorbidity long-term condition review. However, causes of variation in providing and recording of high-quality care still need to be identified and mechanisms need to be explored to ensure appropriate delivery of care to all patients.

**Acknowledgements** The authors thank the OA Research Users' Group, NIHR West Midlands CRN Primary Care, and the network, health informatics, study coordinator, research nurse and administrative staff at Keele University's Arthritis Research UK Primary Care Centre and Keele Clinical Trials Unit for all their support and assistance with this study. The authors give special thanks to all of the staff and patients at the participating general practices, and the GP facilitators who provided support to the general practices involved in the study.

**Contributors** HJ and LAB performed the analysis and drafted and revised the paper. KPJ and JJE developed the analysis plan, cleaned the data, and drafted and revised the paper; KSD is the PI for the study, led the design of the MOSAICS study, and revised the paper; EC, ZP and AGF were involved in the interpretation of the findings and revised the paper. All authors have approved the final version.

**Funding** This paper presents independent research funded by the National Institute for Health Research (NIHR) Programme Grant (RP-PG-0407-10386). The views expressed in this paper are those of the authors and not necessarily those of the NHS, the NIHR or the Department of Health. This research was also funded by the Arthritis Research UK Primary Care Centre grant (Grant No. 18139). HJ and LAB were funded by an NIHR Research Methods Training Fellowship. KSD is partly funded by the NIHR Collaborations for Leadership in Applied Health Research and Care West Midlands and a Knowledge Mobilisation Research Fellowship (KMRF-2014-03-002) from the NIHR. EC and JJE are Academic Clinical Lecturers in Primary Care funded by the NIHR; JJE was previously supported by an In-Practice Fellowship from the NIHR.

**Competing interests** KPJ reports grants from National Institute for Health Research, grants from Arthritis Research UK, during the conduct of the study. KSD reports grants from Arthritis Research UK Centre in Primary Care grant, grants from National Institute for Health Research (NIHR) Programme Grant (RP-PG-0407-10386), during the conduct of the study; grants from Knowledge Mobilisation Research Fellowship (KMRF-2014-03-002), non-financial support from National Institute of Health and Care Excellence, other from Bone and Joint Decade 2015 Conference Oslo, non-financial support from National Institute of Health and Care Excellence Quality Standards, grants from EIT-Health, other from Osteoarthritis Research Society International, outside the submitted work; and Member of the NICE Osteoarthritis Guidelines Development Group CG 59 (2008) and CG 177 (2014). The other authors declare no competing interests.

**Ethics approval** North West Research Ethics Committee, Cheshire (reference: 10/H1017/76).

**Provenance and peer review** Not commissioned; externally peer reviewed.

**Data sharing statement** The Centre has established data sharing arrangements to support joint publications and other research collaborations. Applications for access to anonymised data from our research databases are reviewed by the Centre's Data Custodian and Academic Proposal (DCAP) Committee and a decision regarding access to the data is made subject to the NRES ethical approval first provided for the study and to new analysis being proposed. Further information on our data sharing procedures can be found on the Centre's website (http://www.keele.ac.uk/pchs/publications/datasharingresources/) or by emailing the Centre's data manager (primarycare.datasharing@keele.ac.uk).

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
