## [Reviewer comments · BMJ Open]

ARTICLE DETAILS

TITLE (PROVISIONAL)	Patterns of routine primary care for osteoarthritis in the UK: a cross-sectional electronic health records study
AUTHORS	Jackson, Holly; Barnett, Lauren; Jordan, Kelvin; Dziedzic, Krysia; Cottrell, Elizabeth; Finney, Andrew; Paskins, Zoe; Edwards, John

VERSION 1 – REVIEW

REVIEWER	Paul Kuijer Academic Medical Center Amsterdam the Netherlands https://www.amc.nl/web/Research/Who-is-Who-in-Research/Who-is-Who-in-Research.htm?p=1398
REVIEW RETURNED	05-Oct-2017

GENERAL COMMENTS	Thank you very much for this informative and innovative paper based on cross sectional data: well performed and described. In the Netherlands we are performing a similar study to assess whether knee OA patients have received or have been offered appropriate care before undergoing knee arthroplasty, including the same quality indicators as you use. Unfortunately, we can not use the patient files and do this by self-reports from patients and orthopedic surgeons. So your paper is a help for us. My suggestions are only minor, so I have added them to the pdf of your manuscript and this file is attached. Hopefully my suggestions are clear and of use. One thought based on the study we are performing in the Netherlands: would it be possible for you to do a similar analysis only for hip and/or knee OA patients regarding your outcomes measures or is the power too low? Again thanks for sharing your results and good luck with finishing the RCT. The reviewer also provided a marked copy with additional comments. Please contact the publisher for full details.
--

REVIEWER	Dr Deirdre Hurley University College Dublin, Ireland
REVIEW RETURNED	07-Oct-2017

GENERAL COMMENTS	This large scale observational study provides a profile of achievement of seven quality indicators of care for clinical OA within four clusters. I have a few minor corrections to improve clarity as detailed below: Methods
--

	P6, line 115 - specify the acceptable methods used for assessment of pain and function P6, line 117 - specify how consideration of a referral to physiotherapy was identified Discussion P15, line 314 - Discuss the possible mechanisms to ensure appropriate delivery of care
--	---

VERSION 1 – AUTHOR RESPONSE

REVIEWER 1:

Comment: One thought based on the study we are performing in the Netherlands: would it be possible for you to do a similar analysis only for hip and/or knee OA patients regarding your outcomes measures or is the power too low?

Response: Thank you for this suggestion. Unfortunately, we do not consider this to be feasible due to small numbers of patients within some clusters.

Abstract (objective) “add 'of patients and clinicians”

Response: this has been done. “To determine common patterns of recorded primary care for osteoarthritis (OA), and patient and provider characteristics associated with the quality of recorded care.” (line 24-25)

Abstract (primary objectives) “Please mention a few, like pain, ...”

Response: this has been done. “Achievement of seven quality indicators of care, (pain/function assessment, information provision, exercise/weight advice, analgesics, physiotherapy), recorded through an electronic template or routinely recorded in the electronic healthcare records, were identified for patients aged ≥ 45 years consulting over a six-month period with clinical OA.” (line 34-35)

Abstract (results) “Could you also provide a mean or median over the practices”

Response: this has been done. “1724 patients (median by practice 183) consulted with clinical OA.” (line 41)

Abstract (results) “Could you also provide % per body region or the most important ones?”

Response: Thank you for this suggestion, we recognise the value of this information, however due to word count constraints, we have not been able to add this to the abstract. We have, however, added this to the main results section (lines 180-182).

Abstract (results) “please add also info about pain and function, similarly as done for the cluster 2, 3 and 4”

Response: Again, we recognise the value of this information but due to word count constraints, we have not been able to add this to the abstract. We have reported that patients in the High cluster had a high probability of receiving or being considered for all care processes as detailed in the main results section and within Table 3.

Abstract (results) “add 5 out of 7?”

Response: As the allocation of patients to clusters was based upon probabilities of cluster membership, we consider it better to avoid specifying the number of care processes received since this varied between cluster members.

Abstract (conclusion) “Please start the conclusion with an answer on the question at stake.”

Response: Thank you for this very relevant comment – we have added an initial sentence “Patterns of recorded care for OA fell into four natural clusters.” (Line 50)

Article summary “Please provide more data in the summary, see for instance the start of your discussion, at least the characteristics of the four clusters in terms of patients and clinicians and of may be the 'poor' reporting stats” “please mention at least a few quality indicators” and “I would omit this one: not the main message of your paper if I am correct”

Response: Thank you for these very relevant comments – we have removed the bullet point about unrecorded care processes but added a new point “Four clusters of recorded care were identified: approximately one-third of patients had a high probability of delivery of most care processes whilst another third had a low probability of any such delivery. The remaining patients had a high probability of pain and function assessment but were distinguished by the probability of delivery or consideration of other aspects of care.” (Lines 66-70)

All seven quality indicators are now referred to in the third bullet point “The analysis used some quality indicators of care newly-implemented in practices through an electronic template (pain/function assessment, information provision, exercise/weight advice, analgesics, physiotherapy), which may have increased the recorded quality of care compared to routine practice”.

Methods (p.5) “Could you tell more how and why these eight were selected or participated?”

Response: We have signposted readers to the study protocol paper which explains practice eligibility: “Practice eligibility has been reported elsewhere [6].” (Lines 97-98).

Methods (p.5) “Sorry, I do not know what Read is: please explain...”

Response: Thank you for highlighting this area of potential uncertainty, particularly for international readers. We have added a reference and included an analogy of the international classification of disease codes to provide more clarity: “UK general practice utilises a system of Read codes (similar in principle to the International Classification of Diseases codes) to record symptoms, morbidities, and care processes [7]” (Lines 104-106).

Methods (p.5) “Please refer here to table 1 and then you can omit the last sentence of this paragraph”

Response: Thank you – this has been done: “Outcome measures were the seven indicators of quality of care for OA in general practice recorded in the EHR (Table 1)” (Line 112-13)

Methods (p.7) “You probably mean sex instead of gender: <https://en.wikipedia.org/wiki/Gender> If so, please later also the remaining of the paper.”

Response: General practice records use the patient’s self-assigned gender at registration, not their biological sex, so we prefer to retain the use of ‘gender’ in this context.

Methods (p.7) “What other clinician characteristics were taken into account?”

Response: As described in lines 108-111, we have allocated an index clinician; the only available clinician characteristic was OA work load as described in lines 148-150 and no other data about clinician characteristics were available.

Methods (p.7) "I am no expert but definitely interested. Could you provide an example so I can get more grip of the BIC?"

Response: Thank you – a reference has been added to explain this further should the reader wish to follow this up: "To determine the optimum number of clusters, we considered the Bayes Information Criterion [14] (BIC, whereby the lowest BIC indicated the best model) with the size of each cluster, and the interpretability of the model." (Line 156-159)

Results (p.8) "Could you also provide a mean or median over the practices?"

Response: thank you - this has been done: "During the six-month period, 1724 patients (median per practice n=183) consulted with a recorded clinical OA code and triggered the e-template." (Line 172)

Results (p.8) "Could you also present data about the participating clinicians?"

Response: As above, given our methodology of using electronic health records, additional data about clinician characteristics are not available.

Results (p.9) "Could you be more specific how you weigh these criteria?"

Response: no specific weightings were used for the cluster sizes, goodness-of-fit and interpretability. The model choice was based on the collective interpretation of the team which included experienced statisticians and clinicians. We have added "clinical" to interpretability to be more specific: "Based on the cluster sizes, goodness-of-fit statistics, and clinical interpretability, the four-cluster model was chosen as the optimal model." (Line 187-188)

Results (p.9) "What do you mean by Supplementary?"

Response: Thank you for highlighting this. Unfortunately, the supplementary table was not included in the submission. It has now been added to the main manuscript as Table 4 and the subsequent tables renumbered. Given the number of tables, it could be moved to a supplementary appendix outside the main manuscript if preferred.

Results (p.9) "Please provide a specific example from the table as illustration"

Response: Thank you – for clarity we have now added an example ("For example, in the pain assessment domain, there was no difference between the counts of observed and expected provision for the High and Moderate clusters, and a difference of only one patient in the Low and None clusters; for OA information provision, this was observed more frequently than expected for the High cluster (observed n=620 compared to 613 expected) but less frequently for the Moderate (59 vs. 85) and Low (85 vs. 91) clusters.") (Line 209-214)

Conclusion (p.14) "Also for hand OA?"

Response: Thank you for highlighting this. The National Institute for Health and Care Excellence guidelines, which the MOSAICS study was designed to implement, did not differentiate between site of clinical OA and the need for exercise and weight loss in people who were found to be overweight or

obese. Thus, everyone with a BMI of 35 and over with a record of clinical OA should have been advised about exercise and weight loss if the guidelines were being adhered to.

Table 1 (p.20) “Add Seven quality ...”, “Number these from 1 to 7?”

Response: thank you - this has been done: “Table 1: Seven quality Indicators and categories used for latent class analysis”.

Tables 4 and 5 (p.23-24) “May be indicate in the rows which data refer to patients and which to clinicians”

Response: thank you - this has been done. Please note that the tables have been renumbered due to inclusion of supplementary table 1 (as Table 4) (see renumbered Tables 5 and 6)

REVIEWER 2:

Methods P6, line 115 - specify the acceptable methods used for assessment of pain and function

Response: Thank you - the National Institute for Health and Care Excellence OA management guidelines do not specify a particular method for clinical assessment of pain and function. Within the main trial, clinicians were asked to assess pain and function within a model consultation, however there was no prescriptive way of doing so provided. The Delphi exercise for the development of the model OA consultation identified a wish not to include a specific test for function. Hence, the quality indicators were pragmatically considered to have been achieved if there were any record of pain and function assessment.

P6, line 117 - specify how consideration of a referral to physiotherapy was identified

Response: Thank you – the full methodology of interpretation of the possible responses from the electronic template is detailed in a previous paper. We have modified the manuscript to improve the signposting of the additional information (“The design, interpretation, and effects of the e-template have previously been reported [5]). (Line 124-125)

Discussion

P15, line 314 - Discuss the possible mechanisms to ensure appropriate delivery of care

Response: Thank you. We have revised the final paragraph of the conclusion to read “A lack of a systematic approach to people with OA has previously been reported [27]. A structured annual review for people with OA [28] as recommended by NICE may help. This may possibly be nurse-led and integrated, where appropriate, into a multimorbidity long-term condition review. However, causes of variation in providing and recording of high quality care still need to be identified and mechanisms need to be explored to ensure appropriate delivery of care to all patients. (Lines 324-329)

VERSION 2 – REVIEW

REVIEWER	Paul Kuijer Coronel Institute of Occupational Health, Academic Medical Center, University of Amsterdam, Amsterdam Public Health research institute, Amsterdam The Netherlands
REVIEW RETURNED	27-Nov-2017
GENERAL COMMENTS	Thank you for your addressing my former comments and suggestions in your paper and your reply.

REVIEWER	Dr Deirdre Hurley University College Dublin, Ireland
REVIEW RETURNED	29-Nov-2017

GENERAL COMMENTS	The authors have satisfactorily addressed my comments.
--